# Urinary Continence Recovery after Retzius-Sparing Robot Assisted Radical Prostatectomy and Adjuvant Radiation Therapy

**DOI:** 10.3390/cancers15174390

**Published:** 2023-09-01

**Authors:** Alberto Olivero, Stefano Tappero, Ofir Maltzman, Enrico Vecchio, Giorgia Granelli, Silvia Secco, Alberto Caviglia, Aldo Massimo Bocciardi, Antonio Galfano, Paolo Dell’Oglio

**Affiliations:** 1Department of Urology, ASST Grande Ospedale Metropolitano Niguarda, 20162 Milan, Italy; oliveroalby@gmail.com (A.O.); ofirmal9@gmail.com (O.M.); enrico.vecchio.5@gmail.com (E.V.); giorgia.granelli.gg@gmail.com (G.G.); silviasecc@gmail.com (S.S.); caviglialberto@gmail.com (A.C.); aldobocciardi@gmail.com (A.M.B.); antoniogalfano@gmail.com (A.G.); paolo.delloglio@gmail.com (P.D.); 2IRCCS Ospedale Policlinico San Martino, University of Genova, 16131 Genova, Italy; 3Department of Surgical and Diagnostic Integrated Sciences (DISC), University of Genova, 16131 Genova, Italy; 4Department of Urology, Netherlands Cancer Institute-Antoni Van Leeuwenhoek Hospital, Plesmanlaan 121, 1066 CX Amsterdam, The Netherlands; 5Interventional Molecular Imaging Laboratory, Department of Radiology, Leiden University Medical Center, Rapenburg 70, 2311 EZ Leiden, The Netherlands

**Keywords:** prostate cancer, robot-assisted radical prostatectomy, adjuvant radiation therapy, urinary continence

## Abstract

**Simple Summary:**

The current study is about Retzius Sparing Robot-Assisted Radical Prostatectomy, a surgical approach deemed capable of improving continence recovery after surgery due to the preservation of the structures surrounding the prostate. Specifically, the association between adjuvant Radiation Therapy and urinary continence after prostatectomy was the topic of interest of the study. Patients with prostate cancer treated at a high-volume European institution were analyzed. Based on the results of the study, Adjuvant Radiation Therapy does not significantly undermine urinary continence recovery.

**Abstract:**

Retzius-sparing robot-assisted radical prostatectomy (RS-RARP) allows the preservation of the structures advocated to play a crucial role in the continence mechanism. This study aims to evaluate the association between adjuvant radiation therapy (aRT) and urinary continence (UC) recovery after RS-RARP. For the purpose of the current study, all patients submitted to RS-RARP for prostate cancer (PCa) at a single high-volume European institution between January 2010 and December 2021 were identified. Only patients that harbored pT2 stage with positive surgical margins or pT3/pN1 stage with or without positive surgical margins were included in the analyses. Two groups of patients were identified as follows: patients who had undergone aRT and patients submitted to observation (no-aRT patients). As per definition, aRT was delivered within 1–6 months after surgery. After 1:1 propensity score matching, 124 aRT patients were compared with 124 no-aRT patients who continued standard follow-up protocol after surgery. UC recovery was 81 vs. 84% in aRT vs. no-aRT patients (*p* = 0.7). In multivariable Cox regression analyses, aRT did not reach the independent predictor status for UC recovery at 12 months. In the subgroup analysis including only aRT patients, only the nerve-sparing technique was independently associated with UC recovery at 12 months. Conversely, the type of aRT (IMRT/VMAT vs. 3D-CRT) did not reach the independent predictor status for UC recovery at 12 months. The current study is the first to address the association between aRT and UC recovery in patients treated with RS-RARP for PCa. Based on our data, aRT is not associated with worse UC recovery. In the cohort of patients treated with aRT, the nerve-sparing technique independently predicted UC recovery.

## 1. Introduction

Radical prostatectomy represents an effective treatment for organ-confined prostate cancer (PCa) patients [1]. Although it is associated with excellent long-term cancer control [2], additional postoperative treatments should be considered in patients with adverse pathological findings. Randomized controlled trials [3,4,5] failed to observe a benefit of adjuvant radiotherapy (aRT) in terms of progression-free survival (PFS) compared with early-salvage radiotherapy (sRT). However, due to the small number of patients with adverse pathology (i.e., International Society of Urological Pathology (ISUP) grade 4–5, ≥pT3 with or without positive surgical margins (PSM)) included in these trials, aRT remains a recommended treatment option in highly selected patients with adverse pathology [6]. Gastrointestinal and genito-urinary side effects during and after RT are common and represent a major issue [1,7]. In this regard, the potential detrimental effect of aRT on urinary continence (UC) recovery significantly impairs long-term quality of life [8].

To date, few studies evaluated the impact of aRT on UC recovery. All the reports are limited by small sample size [9,10] and are based on noncontemporary patients [9,10,11,12]. Moreover, the majority of these studies did not account for all parameters associated with UC recovery, such as tumor characteristics, patient comorbidity [13], previous benign prostatic hyperplasia (BPH) surgery, surgical technique (i.e., nerve-sparing vs. non-nerve-sparing [14]), and type of adjuvant radiotherapy (i.e. three-dimensional conformal RT (3D-CRT) vs. intensity-modulated RT (IMRT) or volumetric arc radiation therapy (VMAT) with image-guided RT) [15,16,17,18]. Furthermore, none of these studies exclusively focused on patients treated with Retzius-sparing robot-assisted radical prostatectomy (RS-RARP). RS-RARP is a valid surgical treatment option for PCa patients according to the European Association of Urology guidelines [1,19,20,21,22]. This has recently been confirmed even in the specific scenario of D’Amico high-risk PCa patients [23,24].

We addressed this knowledge gap, and we tested the association between aRT and UC recovery after RS-RARP. We hypothesized that aRT might not significantly impact UC recovery after surgery. The rationale of such hypothesis stems from the preservation of all the structures that are crucial in the mechanism of UC (e.g., endopelvic fascia, puboprostatic ligaments, and Santorini plexus), which is allowed by the RS-RARP technique. We relied on a large cohort of PCa patients treated with RS-RARP at a single high-volume European institution (2010–2021).

## 2. Materials and Methods

### 2.1. Population

The current study is a retrospective analysis of 2475 PCa patients submitted to RS-RARP with or without pelvic lymph node dissection (PLND) at a single high-volume European institution (ASST Grande Ospedale Metropolitano Niguarda, Milan, Italy) between January 2010 and December 2021. All RS-RARP procedures were performed with a four-arm da Vinci Si Surgical System (Intuitive Surgical, Sunnyvale, CA, USA) with a transperitoneal approach, as previously described [19], by five experienced robotic surgeons. The indication for mono- or bilateral neuro-vascular bundle preservation (nerve-sparing—NS) was based on disease characteristics at diagnosis and the treating physician’s clinical judgment, regardless of preoperative erectile status.

The inclusion criteria for the study were pT2 stage with PSM or pT3/pN1 stage with or without PSM. We divided the patients into two groups: patients who received aRT and patients who underwent observation. Patients with incomplete/missing postoperative RT data (*n* = 207), incomplete follow-up data (*n* = 482), and missing pre- and intraoperative data (*n* = 102) were not included in the study. No patient received neoadjuvant, adjuvant, or salvage hormonal therapy during the study period. No patient in the non-aRT group received salvage RT within the study period. Overall, 1684 assessable patients were identified. Of those, 124 patients (8.6%) received aRT.

### 2.2. Evaluated Variables and Study End Points

For each patient, the following variables were collected: age at surgery, body mass index (BMI), Charlson comorbidity index (CCI), previous BPH surgery, D’Amico risk group [25], pathological T and N stages, and positive surgical margins (PSM). All patients were preoperatively continent (i.e., no urinary pads and no bladder catheter) and had complete data on postoperative continence.

Administration of aRT was based on the indication given by each treating physician and followed an extensive discussion with patients about treatment options and expectations. Adjuvant RT was delivered within 1–6 months after surgery. Between 2010 and 2014, all patients underwent three-dimensional conformal radiation therapy (3D-CRT). Since 2015, intensity-modulated RT (IMRT) with image-guided RT or volumetric arc radiation therapy (VMAT) with image-guided RT has been used. For each patient submitted to aRT, data concerning radiation dose (Gy) and time interval between RS-RARP and aRT were collected.

The end point of this study was to address the association between aRT after RS-RARP and UC recovery. Urinary continence recovery was defined as the use of zero or one safety pad per day at last follow-up, as previously conducted [24,26]. Patients were evaluated at 1, 3, 6, and 12 months postoperatively and every 6 months thereafter. At each visit, postoperative UC recovery was assessed.

### 2.3. Statistical Analysis

Established recommendations for statistical analyses, reporting, and interpretation of the results were applied [27]. The analysis consisted of four steps. First, in order to provide the most unbiased comparison between aRT vs. no-aRT patients, 1:1 propensity score matching was performed. Matching variables consisted of CCI, D’Amico risk group, and NS technique. Second, Kaplan–Meier plots graphically depicted 12-month UC recovery rates according to the treatment type (aRT vs. no-aRT). Third, univariable and multivariable Cox regression analyses tested the association between aRT and UC recovery in the overall population. Covariates included age at surgery, BMI, CCI, year of treatment, previous BPH surgery, pathological status (pT2 with PSM vs. pT3 with or without PSM vs. pTany-pN1 with or without PSM), NS technique, and surgical experience (SE). SE was coded as the number of previous RS-RARP performed by each surgeon at the time of the index patient’s operation, as previously conducted [26,28]. Fourth and last, the above-described methodology was reapplied in a sensitivity analysis, specifically testing for independent predictors of UC recovery in the subgroup of aRT patients. Here, covariates consisted of type of aRT (3D-CRT vs. IMRT/VMAT), radiation dose (Gy), time interval between RS-RARP and aRT, year of treatment, age at RS-RARP, NS technique, and pathological status. In all multivariable models, the number of covariates met the criteria for model overfitting prevention. In all statistical analyses, the R software environment for statistical computing and graphics (R version 4.1.2; R Foundation for Statistical Computing, Vienna, Austria) was used. All tests were two-sided, with a level of significance set at *p* < 0.05.

## 3. Results

After 1:1 propensity score matching, 124 aRT patients were compared with 124 no-aRT counterparts. The median follow-up was 49 months. According to D’Amico risk stratification, 88 vs. 81% of, respectively, aRT vs. no-aRT patients were classified as high-risk (Table 1). Compared with no-aRT patients, those who received aRT harbored worse pathological characteristics (pT and pN stages, pathologic ISUP grade; Table 1). Partial and full NS was performed in 9 vs. 9% and 34 vs. 35%, respectively, of aRT vs. no-aRT patients (*p* = 0.9; Table 1). No difference was recorded in terms of PSM (60 vs. 63%, *p* = 0.6; Table 1). At 12 months from RS-RARP, UC recovery rates were 81 vs. 84% in aRT vs. no-aRT patients (log-rank *p* = 0.9; Figure 1).

In univariable Cox regression analyses, aRT was associated with worse UC recovery at 12 months (hazard ratio [HR]: 0.87; 95% confidence interval [CI]: 0.73–0.97; *p* = 0.040; Table 2). Such association was not confirmed in multivariable Cox regression model (Table 2). In the sensitivity analysis focused on aRT patients, only the NS technique was independently associated with UC recovery at 12 months (partial NS, HR: 1.79; 95% CI: 1.01–3.14; *p* = 0.045; full NS, HR: 2.24; 95% CI: 1.17–5.21; *p* = 0.032; Table 3). Conversely, the type of aRT did not reach the independent predictor status (HR 1.28; 95% CI: 0.72–2.27; *p* = 0.4).

## 4. Discussion

Adjuvant RT remains a recommended treatment option in a selected group of patients with high-risk diseases and adverse pathology. However, the real aRT advantage regarding oncological outcomes is still debated: three randomized trials [3,4,29] and a meta-analysis [30] found no difference in progression-free survival for adjuvant compared with early salvage RT. In contrast, a recent study from Tilki and colleagues showed a reduction in all-cause mortality risk for patients who have undergone aRT with pathological adverse features [6].

While the IMRT effect on UC is well described [15], very few studies analyzed functional outcomes of aRT after radical prostatectomy, and they all reported the potential detrimental aRT effect on UC [10,11,12]. None of these studies exclusively focused on RS-RARP.

The current study aimed to address this knowledge gap, addressing the association between aRT after RS-RARP and UC recovery, relying on a large cohort of PCa patients treated with RS-RARP at a single high-volume European institution.

Our results yield several noteworthy observations.

First, we reported an overall rate of 12 months UC recovery of 81 vs. 84% in aRT vs. no-aRT patients. These patients, according to the EAU risk group, are more at risk of failing UC recovery; obtaining a high continence rate in this subset underlines the value of RS technique on UC recovery [20,24,31].

Second, in this study, aRT was associated with worse UC recovery only in the univariable model. When we accounted for multiple confounders, we failed to observe an association between aRT and the outcome of interest, suggesting that UC recovery after RS-RARP might not be undermined by the delivery of aRT. Our findings are in contrast with the previous literature on the standard RARP. For example, Suardi and colleagues reported a detrimental effect on UC recovery and on time to UC recovery in patients treated with standard RP (HR: 0.57 95% IC:0.43–0.76 *p*-value < 0.001) [11]. Zaffuto and colleagues reported a 3-year UC recovery rate of 70.7%, 59.0%, and 42.2% in patients who received no radiotherapy, salvage radiotherapy, or adjuvant radiotherapy, respectively (*p* < 0.001) [12], and these results were confirmed in another report [10]. Our results could lead to some speculative hypotheses: in the posterior approach (i.e., RS-RARP), the dissection starts posteriorly at the pouch of Douglas, first dissecting the seminal vesicles and progressing caudally behind the prostate. This allows to spare all the anterior support structures of bladder and prostate that are resected in the anterior approach. If it has a role in UC recovery after surgery, it should also have one in UC maintenance after aRT.

Third, when we performed a sensitivity analysis and exclusively focused on aRT patients, the type of aRT (3D-CRT vs. IMRT/VMAT) failed to reach the independent predictor status. Again, these findings might be explained by the preservation of the structures of the Retzius space that are crucial in the continence mechanism and guarantee excellent functional outcomes, regardless of aRT type. However, these findings need to be interpreted with caution, considering the retrospective nature of the study and the relatively short follow-up of our cohort.

Of note, in the sensitivity analysis, only the NS technique was independently associated with UC recovery at 12 months. These findings are in line with our previously reported data [32]. However, the advantage of a conservative approach in this setting of patients with aggressive disease should always be weighted with the risk of PSM and poor cancer control.

Last but not least, in contrast to our previously reported data [32], we found that age at surgery did not predict 12-month UC recovery, probably because the detrimental effect on the tissue of radiation nullified the advantages of young age before the surgery.

Our results must be interpreted considering some limitations. First and foremost, the current study relied on the analyses of retrospectively collected data. Therefore, the reported results must be interpreted within the boundaries of such limitation. Second, despite the best attempts at multivariable adjustment, differences between the study cohorts might not be completely accounted for. However, this limitation is shared with all the other existing retrospective clinical studies and cannot be overcome. Third and last, our analyses did not rely on validated questionnaires on symptomatologic bother and quality of life.

Contrariwise, our study has several points of strength. It is the first-ever published study to extensively investigate UC recovery in the specific setting of aRT after RS-RARP. We reported the radiation dose administered, and this could lead to comparison studies in the future. In addition, we overcame the potential bias of the single operator ability by including different surgeons.

## 5. Conclusions

RS-RARP provides excellent functional outcomes. This is true even in patients with the worst preoperative and postoperative features, among whom those potentially requiring adjuvant treatments (e.g., PSMs, pT3, and pN1). In the current study, we did not find significant differences in 12-month UC recovery between RS-RARP patients treated and nontreated with aRT, and aRT did not reach the independent predictor status for UC recovery.

## Figures and Tables

**Figure 1 cancers-15-04390-f001:**
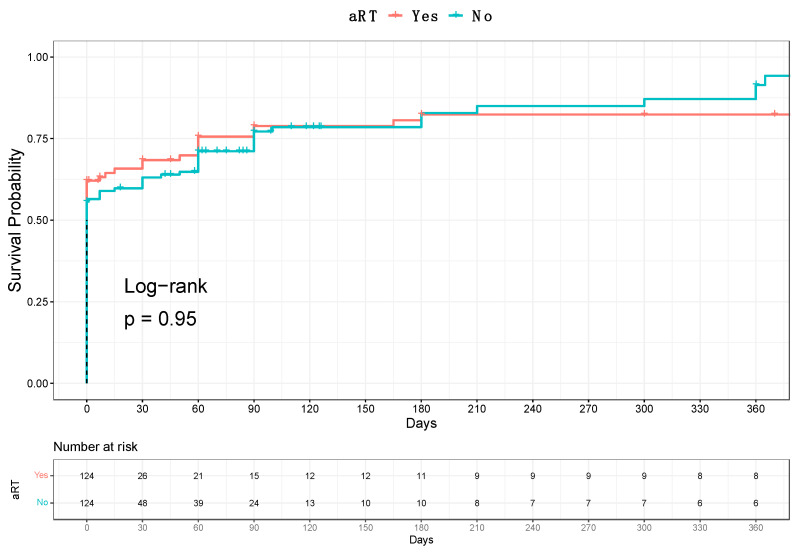
Kaplan-Meier plot depicting urinary continence recovery at 12 months from RS-RARP according to adjuvant radiotherapy status.

**Table 1 cancers-15-04390-t001:** Clinical and pathological characteristics of 248 patients treated with RS-RARP having harbored pT2 stage with positive surgical margins or pT3/pN1 stage with or without positive surgical margins, according to adjuvant radiotherapy delivery, after 1:1 propensity score matching. Matching variables: Charlson comorbidity index, D’Amico risk classification groups, and nerve-sparing technique.

Characteristic	Overall, *n* = 248 ^1^	aRT, *n* = 124 ^1^	No aRT, *n* = 124 ^1^	*p*-Value ^2^
**Year of treatment**				0.10
2010–2015	172 (69%)	80 (65%)	92 (74%)	
2016–2021	76 (31%)	44 (35%)	32 (26%)	
**Age at RS-RARP**	67 (62, 71)	67 (60, 71)	68 (63, 72)	0.1
**Body mass index, kg/mq**	26 (24, 28)	26 (24, 28)	26 (24, 28)	0.9
**Charlson comorbidity index**				0.3
2	36 (15%)	22 (18%)	14 (11%)	
3	94 (38%)	48 (39%)	46 (37%)	
≥4	118 (48%)	54 (44%)	64 (52%)	
**Previous surgery for BPH**	16 (7%)	6 (5%)	10 (8%)	0.3
**D’Amico high-risk group**	210 (85%)	109 (88%)	101 (81%)	0.4
**Pathological T stage ≥ 3**	209 (84%)	116 (94%)	93 (75%)	**<0.001**
**Pathological N stage 1**	71 (29%)	65 (52%)	6 (5%)	**<0.001**
**ISUP grade at RS-RARP 4–5**	83 (33%)	64 (52%)	19 (15%)	**<0.001**
**Positive surgical margins**	152 (61%)	74 (60%)	78 (63%)	0.6
**Patient category**				**<0.001**
pT2 with positive surgical margins	34 (14%)	4 (3%)	30 (24%)	
pT3 with or without positive surgical margins	143 (58%)	55 (44%)	88 (71%)	
pTany-pN1	71 (29%)	65 (52%)	6 (5%)	
**Nerve-sparing technique**				0.9
No nerve-sparing	140 (56%)	71 (57%)	69 (56%)	
Full nerve-sparing	86 (35%)	42 (34%)	44 (35%)	
Partial nerve-sparing	22 (9%)	11 (9%)	11 (9%)	
**Surgical experience ≥ 100**	229 (92%)	111 (90%)	118 (95%)	0.1
**Urinary continence recovery status at last follow-up**				0.7
0–1 pad per-day	204 (82%)	100 (81%)	104 (84%)	
>1 pad per-day	44 (18%)	24 (19%)	20 (16%)	
**Time from RS-RARP to aRT, months**	-	4 (3–5)	-	
**Radiation dose, Gy**	-	70 (68, 74)	-	
**Type of aRT**				
3D-CRT	-	67 (54%)	-	
IMRT/VMAT	-	57 (46%)	-	

^1^ Median (IQR); *n* (%); ^2^ Wilcoxon rank sum test; Fisher’s exact test; Pearson’s chi-squared test. Legend: RS-RARP: Retzius-sparing robot-assisted radical prostatectomy; BPH: benign prostate hyperplasia; ISUP: International Society of Urological Pathology; aRT: adjuvant radiation therapy; 3D-CRT: three-dimensional conformal radiation therapy); IMRT: Intensity Modulated Radiation Therapy; VMAT: Volumetric Modulated Arc Therapy. Bold characters are for statistically significant *p* values.

**Table 2 cancers-15-04390-t002:** Cox regression models testing urinary continence recovery at 12 months from RS-RARP.

	Univariable			Multivariable		
Characteristic	HR ^1^	95% CI ^1^	*p*-Value	HR ^1^	95% CI ^1^	*p*-Value
**Adjuvant radiotherapy, aRT**						
No aRT	—	—		—	—	
aRT	0.87	0.73, 0.97	**0.040**	0.65	0.32, 1.29	0.2
**Age at RS-RARP**	0.96	0.94, 0.98	**<0.001**	0.98	0.88, 1.09	0.7
**Body mass index**	1.01	0.95, 1.08	0.65	1.02	0.95, 1.09	0.6
**Charlson comorbidity index**	0.80	0.69, 0.93	**0.004**	1.09	0.72, 1.66	0.7
**Year of treatment**	0.99	0.94, 1.03	0.5	0.98	0.88, 1.09	0.7
**Previous surgery for BPH**	0.64	0.35, 1.19	0.2	1.17	0.38, 3.58	0.8
**Pathological status**						
pT2 with PSM	—	—		—	—	
pT3 with or without PSM	0.81	0.54, 1.23	0.3	1.39	0.60, 3.25	0.4
pN1 with or without PSM	0.70	0.44, 1.09	0.1	1.30	0.51, 3.30	0.6
**Nerve-sparing technique**						
No nerve-sparing	—	—		—	—	
Partial nerve-sparing	1.34	1.00, 1.59	0.058	1.53	0.69, 3.88	0.3
Full nerve-sparing	1.44	1.08, 1.86	**0.004**	0.96	0.46, 1.98	0.9
**Surgical experience**						
Less than 100	—	—		—	—	
More than 100	0.84	0.52, 1.34	0.5	1.02	0.49, 2.12	0.9

^1^ HR: Hazard ratio; CI: confidence interval. Legend. RS-RARP: Retzius-sparing robot-assisted radical prostatectomy; aRT: adjuvant radiotherapy; BPH: benign prostate hyperplasia; PSM: positive surgical margins.

**Table 3 cancers-15-04390-t003:** Cox regression models testing urinary continence recovery at 12 months from RS-RARP, in the subset of aRT patients.

	Univariable			Multivariable		
Characteristic	HR ^1^	95% CI ^1^	*p*-Value	HR ^1^	95% CI ^1^	*p*-Value
**Type of aRT**						
3D-CRT	—	—		—	—	
IMRT/VMAT	0.85	0.57, 1.26	0.4	1.28	0.72, 2.27	0.4
**Radiation dose, Gy**	0.99	0.98, 1.00	**0.023**	0.99	0.98, 1.01	0.3
**Time from RS-RARP to aRT**	1.01	0.83, 1.22	0.9	0.89	0.71, 1.12	0.3
**Year of treatment**	0.95	0.87, 1.03	0.2	0.87	0.69, 1.09	0.2
**Age at RS-RARP**	0.97	0.94, 0.99	**0.011**	0.98	0.94, 1.01	0.2
**Nerve-sparing technique**						
No nerve-sparing	—	—		—	—	
Partial nerve-sparing	1.31	0.85, 2.01	0.23	1.79	1.01, 3.14	**0.045**
Full nerve-sparing	1.51	0.76, 2.97	0.24	2.24	1.17, 5.21	**0.032**
**Pathological status**						
pT2 with PSM	—	—		—	—	
pT3 with or without PSM	1.79	0.55, 5.80	0.3	0.29	0.05, 3.06	0.3
pN1 with or without PSM	1.23	0.38, 3.97	0.7	0.18	0.03, 2.05	0.2

^1^ HR: Hazard ratio; CI: confidence interval. Legend. RS-RARP: Retzius-sparing robot-assisted radical prostatectomy; BPH: benign prostate hyperplasia; aRT: adjuvant radiation therapy; 3D-CRT: Three-Dimensional Conformal Radiation Therapy; IMRT: intensity-modulated radiation therapy; VMAT: volumetric-modulated arc therapy; PSM: positive surgical margins.

## Data Availability

All data generated for this analysis were from anonymized database. The code for the analyses will be made available upon request.

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
