# Peer review of "Urinary Continence Recovery after Retzius-Sparing Robot Assisted Radical Prostatectomy and Adjuvant Radiation Therapy"

_cancers, 2023, doi:10.3390/cancers15174390_

Round 1

Reviewer 1 Report

The authors retrospectively evaluated urinary continence recovery after RS-RARP who received adjuvant RT.

They concluded that aRT did not affect urinary continence recovery using PSM compared to non-RT group in their Institution.

The idea is interesting and important for consideration of controlling high risk Pca who received RARP.  But the reviewer has some questions and requests to the authors to improve their article.

1)     The authors should describe a definition of urinary continence.  This is critical and mandatory.

2)     The authors should present urinary continence rate of all patients who received RS-RARP in their Institution.  We want to know whether the groups presented is better or worse in UC recovery compared to general patients with RS-RARP.

3)     I cannot see supplementary data of Kaplan-Meier curve of urinary continence recovery.  The reviewer recommends presenting it by the full Figure but not by the supplementary.

Author Response

The authors retrospectively evaluated urinary continence recovery after RS-RARP who received adjuvant RT. They concluded that aRT did not affect urinary continence recovery using PSM compared to non-RT group in their Institution. The idea is interesting and important for consideration of controlling high risk Pca who received RARP.  But the reviewer has some questions and requests to the authors to improve their article.

1). The authors should describe a definition of urinary continence.  This is critical and mandatory.

Response: We thank the Reviewer for the comment. We modified the manuscript accordingly and provided the definition of urinary continence within the text. The following sentence was added:

Materials and methods, page 3, section Evaluated variables and study end points: “Urinary continence recovery was defined as the use of zero or one safety pad per day at last follow-up, as previously done (24,28)”.

We hope that the Reviewer will consider all the modifications satisfactory.

2). The authors should present urinary continence rate of all patients who received RS-RARP in their Institution. We want to know whether the groups presented is better or worse in UC recovery compared to general patients with RS-RARP.

Response: We thank the Reviewer for the insightful comment. In the current series, we reported a one-year urinary continence recovery of 81 and 84% for patients respectively undergone Retzius-sparing prostatectomy followed and non-followed by adjuvant radiotherapy. These patients harbored unfavorable preoperative tumor characteristics (D’Amico high-risk: 85%) and received partial or full nerve-sparing only in less than one case out of two (No nerve-sparing: 56%).

Conversely, in the overall PCa patient population treated at ASST Niguarda Hospital of Milan (2010-2021), with more diluted unfavorable tumor characteristics, the one-year urinary continence recovery rate reaches 93%. Since these overall data have been recently published (Reference n. 28 in the current manuscript), and since they rely on substantially different patients than those addressed in the current study (i.e., pT2 with surgical margins; pT3; pN1), we decided not to discuss these results within the current manuscript. We hope that the Reviewer will consider our explanations satisfactory.

3). I cannot see supplementary data of Kaplan-Meier curve of urinary continence recovery.  The reviewer recommends presenting it by the full Figure but not by the supplementary.

Response: We thank the Reviewer for the comment. We will re-submit the Figure 1 not as supplementary, but as integrant part of the manuscript.

Reviewer 2 Report

Dear Authors,

I reviewed with interest the paper entitled “Urinary Continence Recovery After Retzius-Sparing Robot Assisted Radical Prostatectomy and Adjuvant Radiation Therapy”.

First, I would strongly congratulate with the authors for their hard work for the present study, which covers an actual and very interesting topic, such as functional outcomes after treatment for prostate cancer (PCa). Specifically, Authors aimed to retrospectively evaluate the association between adjuvant Radiation Therapy and urinary continence recovery after Retzius-Sparing Robot Assisted Radical Prostatectomy (RS-RARP).

I found the present study interesting and well written - no major concerns with language editing.

The title is clear and descriptive of what authors have explored in their work.

The Introduction provides a background which is relevant to the study, it is clearly stated and exhaustive. Tables are clear and not repetitive, as well as the Results - which are interesting and significant. Methods are clearly described and in enough detail. Statistical assessment is well conducted and the paper results methodologically correct. Discussion is adequately presented, and interpretations and conclusions are well stated and justified by results. I have not further comments.

Author Response

We are truly honored and thank the Reviewer for her/his positive comments.

Round 2

Reviewer 1 Report

The authors responded appropriately to my comments.